# Intraoperative Assessment of Tumor Margins in Tissue Sections with Hyperspectral Imaging and Machine Learning

**DOI:** 10.3390/cancers15010213

**Published:** 2022-12-29

**Authors:** David Pertzborn, Hoang-Ngan Nguyen, Katharina Hüttmann, Jonas Prengel, Günther Ernst, Orlando Guntinas-Lichius, Ferdinand von Eggeling, Franziska Hoffmann

**Affiliations:** Innovative Biophotonics & MALDI Imaging, ENT Department, Jena University Hospital, 07747 Jena, Germany

**Keywords:** oral squamous cell carcinoma, hyperspectral imaging, machine learning, intraoperative assessment, tumor detection, tumor margins

## Abstract

**Simple Summary:**

The complete resection of the malignant tumor during surgery is crucial for the patient’s survival. To date, surgeons have been intraoperatively supported by information from a pathologist, who performs a frozen section analysis of resected tissue. This tumor margin evaluation is subjective, methodologically limited and underlies a selection bias. Hyperspectral imaging (HSI) is an established and rapid supporting technique. New artificial-intelligence-based techniques such as machine learning (ML) can harness this complex spectral information for the verification of cancer tissue. We performed HSI on 23 unstained tissue sections from seven patients with oral squamous cell carcinoma and trained the ML model for tumor recognition resulting in an accuracy of 0.76, a specificity of 0.89 and a sensitivity of 0.48. The results were in accordance with the histopathological annotations and do, therefore, enable the delineation of tumor margins with high speed and accuracy during surgery.

**Abstract:**

The intraoperative assessment of tumor margins of head and neck cancer is crucial for complete tumor resection and patient outcome. The current standard is to take tumor biopsies during surgery for frozen section analysis by a pathologist after H&E staining. This evaluation is time-consuming, subjective, methodologically limited and underlies a selection bias. Optical methods such as hyperspectral imaging (HSI) are therefore of high interest to overcome these limitations. We aimed to analyze the feasibility and accuracy of an intraoperative HSI assessment on unstained tissue sections taken from seven patients with oral squamous cell carcinoma. Afterwards, the tissue sections were subjected to standard histopathological processing and evaluation. We trained different machine learning models on the HSI data, including a supervised 3D convolutional neural network to perform tumor detection. The results were congruent with the histopathological annotations. Therefore, this approach enables the delineation of tumor margins with artificial HSI-based histopathological information during surgery with high speed and accuracy on par with traditional intraoperative tumor margin assessment (Accuracy: 0.76, Specificity: 0.89, Sensitivity: 0.48). With this, we introduce HSI in combination with ML hyperspectral imaging as a potential new tool for intraoperative tumor margin assessment.

## 1. Introduction

Surgery is a cornerstone of the standard therapy of patients with oral squamous cell carcinoma (OSCC). The complete resection (R0 status) has a high impact on patient outcome. Survival of patients with incomplete tumor resection is significantly lower [1,2,3]. The main challenge is a reliable intraoperative asessment the complete resection of the tumor. Currently used intraoperative in vivo imaging devices are lacking the ability to clearly define the tumor margins. The current standard of care for intraoperative tumor margin assessment is the frozen section analysis of biopsies taken from the resection margins. This intraoperative histopathological evaluation is highly subjective with an overall accuracy of only 71.3% [4,5]. Therefore, an objective and fast application and workflow are needed to achieve adequate tumor margin detection.

Hyperspectral imaging (HSI) is an optical method to detect wavelengths beyond the visible spectrum. Light interacts with biological tissues. Different molecular compositions of tumor cells in contrast to non-tumor cells result in different absorption peaks. These can be used as spectral fingerprints for tumor cells. The implementation of HSI makes it possible to enlarge microscopic RGB images by additional channels up to the near-infrared spectral region [6]. Summarized by Ortega et al. [7] it could be shown that the additional information is effective in tumor detection on whole sample specimen.

Published studies for HSI-based segmentation of OSCC and other tumor entities focus on a general assessment of the presence or absence of tumor tissue [8,9,10]. In all these approaches the surface of a bulk biopsy was measured. An exact correlation of HSI and histopathological data has not been performed to date. Other studies that performed HSI measurements on histological slides were either exclusively conducted on already stained sections [11,12,13,14] or for clinical questions other than OSCC tumor detection, e.g., Alzheimer’s disease, nerve lesion detection, thyroid tumor or breast tumor detection [15,16,17,18].

The high amount of data acquired with microscopic HSI makes manual interpretation of the results challenging if not impossible and automated processing a necessity. Machine learning (ML) and digital pathology are the next big step in the field of pathology [19]. The most successful modern ML applications are based on so-called deep learning [20]. The name is derived from how these algorithms are built by stacking multiple hidden layers to solve a multitude of problems, both in the medical field as well as in other areas. Common medical tasks include tumor classification and segmentation [21,22]. Additionally, in some of them, such as breast cancer metastasis detection, machine-learning-based approaches already rival or outperform human experts [23].

To the best of our knowledge, no study has applied machine learning on hyperspectral or traditional RGB images of fresh-frozen unstained tissue sections for interoperative tumor margin assessment. Even for stained fresh-frozen tissue sections, there is a lack of available, annotated data. Therefore, there are few machine-learning-based solutions [24].

This study investigated the potential use of HSI to address the limitations of the rapid intraoperative H&E-based assessment. More precisely, we aim to perform an automated detection of tumor tissue at microscopic resolution using fresh-frozen unstained tissue slices. The detected wavelengths covered the range from 500 to 1000 nm. In this range 100 channels were recorded for further analysis. We performed whole-slide scans with a blank acquisition time of 20 min per cm^2^. The data used are data cubes containing HSI spectra combined with high resolution brightfield scans of the corresponding H&E stained section. We used unsupervised algorithms for data visualization and dimensionality reduction and employed a supervised 3D convolutional neural network (CNN) for tumor classification and segmentation. The results were compared with manual histopathological annotation performed by two experienced pathologists.

The high amount of data acquired with microscopic hyperspectral imaging makes manual interpretation of the results challenging if not impossible and automated processing a necessity.

## 2. Materials and Methods

### 2.1. Clinical Samples

We used tissue samples from seven patients with OSCC. The study was approved by the ethics committee of the Jena University Hospital (No. 2018-1166-material). Written informed consent was obtained from all patients. Five patients were male, and two were female. All samples originated from the oropharyngeal area. After resection, all samples were directly snap-frozen with liquid nitrogen and stored at −80 °C until measurement. Tissue samples were cut into sections of 12 μm thickness with a cryotome. Every 120 μm, a section was mounted on a glass slide for each tissue sample. This led to two to six sections per patient.

### 2.2. HSI Setup and Measurements

All images were captured using an inverted light microscope system (Axio Scope A1; Carl Zeiss AG, Jena, Germany) with a 10× lens (EC Plan Neofluar 10×/0.30 M27; Carl Zeiss AG). RGB images were captured with a CCD-sensor camera (Axiocam MRc; Carl Zeiss AG) with an optical resolution of 1388 × 1040 pixels. HSI images were captured using an integrated HSI platform (TIVITA Mini; Diaspective Vision GmbH, Am Salzhaff, Germany) with its camera connected to the microscope system. The HSI-camera system has an optical resolution of 720 × 540 pixels and a spectral resolution ranging from 500 nm to 1000 nm in 10 nm steps. To calibrate the HSI camera, an HSI datacube was captured from a blank slide with the microscope lamp turned on as a white reference. The illumination intensity of the used halogen lamp was manually adjusted to 95% of the HSI-sensor maximum. An HSI datacube was captured from a blank slide with the microscope lamp turned off as a dark/black reference. Both cameras, the CCD-sensor camera and the HSI camera, were connected to the microscope via a T2-C 1.0× microscope camera adapter.

Due to its size, we systematically scanned the tissue sections completely with the microscope in meandering positions with an overlap of 15%. ZEN blue 2012 edition imaging software (Carl Zeiss AG) was used to systematically scan and acquire microscope imaging positions. RGB images were captured on each imaging position. Using the same imaging position provided by the ZEN imaging software, corresponding HSI images were then captured for each tissue section. The measurement per hyperspectral image took 4 s and the total acquisition time including stage movement took around 5 to 15 min per sample according to their size.

### 2.3. Histopathological Annotation

We stained the tissue sections with hematoxylin and eosin (H&E) after HSI measurements. Histopathological annotation was performed manually by two experienced pathologists. The annotated H&E images were then co-registered with the corresponding RGB images and the HSI datacubes.

### 2.4. Data Preprocessing

We filtered the HSI datacubes for measurement artifacts that appear in the individual pixel in some of the images. Additionally, we removed all samples that contain either no tissue at all, tissue that could not definitely be classified as healthy or tumorous, and samples that contain more than 75% background. We performed no further preprocessing. In particular, we did not perform any type of normalization, since normalization leads to a classification based purely on the shape of the spectra while we assume that there are also intensity differences in the spectra of different regions [25]. In total, we acquired 3591 hyperspectral images, of which 2194 images could be used for our study.

We applied a sixfold cross-validation, always omitting one random slide of five randomly chosen patients. We chose six folds since the highest number of samples for one patient is six, which allows us to use each slide at least once in the validation set. The same sample split was used for all following experiments. We chose to leave out individual sections instead of patients to keep the size of the training and validations set similar between different folds. For the deep learning task of tumor classification, we treated each individual HSI datacube as independent measurement with a single label. A more detailed overview of the data structure is given in Appendix A.

### 2.5. Visualization as False Color Images for Tumor Classification

To visually inspect the hyperspectral datacubes, we performed an unsupervised nonlinear dimensionality reduction using uniform manifold approximation and projection (Umap, Vancouver, BC, Canada) [26]. We trained the dimensionality reduction algorithm on the first 200 non-background spectra of each validation image with the following settings number of components: 3, number of neighbors: 50, minimum distance: 0.5 a spread of 2 and the l2 metric. These parameters are partially based on results published in [27] and we implemented the algorithm using GPU acceleration as described in [28].

### 2.6. Machine-Learning-Based Classification

#### 2.6.1. Linear Support Vector Machine Classifier (SVM)

As a baseline we trained a linear support vector machine classifier (SVM) using statistical gradient descent on each individual spectrum without taking morphological information into account [29]. SVMs have seen wide-spread use for hyperspectral image classification tasks [30] and are in theory well suited. We used the implementation given in [30] with the corresponding parameters.

#### 2.6.2. Fine Tuning on False Color Images and RGB-Microscopy Images

Additionally, we fine-tuned a traditional ResNet architecture for image classification, pre-trained on the Imagenet dataset [31], on the 3 channel outputs of the dimensionality reduction and the traditional RGB images taken with the microscope camera. We utilized label smoothing [32] and data augmentation based on Trivial Augment [33]. On each fold we trained for 30 epochs using the Adam optimizer [34] with a learning rate of 0.001.

#### 2.6.3. Using a 3D Convolutional Neural Network for Classification

We implemented an adapted version of the 3D CNN shown in [35], which is based on a deep residual network (ResNet) [36]. We utilized label smoothing and performed data augmentation in the form of random horizontal or vertical flips of the image, randomly applying gaussian blur as well as randomly modifying the brightness of the hyperspectral image. We trained for 45 epochs using the Adam optimizer with a learning rate of 0.001.

## 3. Results

### 3.1. Traditional Machine Learning for Spectral-Based Tumor Classification

The linear SVM based on each individual spectrum yet lacking additional information about the morphology of the tissue was not able to distinguish healthy from tumorous tissue. This led to either classifying 100% of the samples as healthy or classifying 100% of the samples as tumor. Further analysis shows that the individual intensities differ more between samples than between different tissue types, due to the manual setting of illumination strength. This inhomogeneity in the data and the lack of a clear correlation between tissue types and measured spectra lead us to explore more complex algorithms such as deep learning for this classification task. We show the average spectra for healthy and tumor tissue in Figure 1.

### 3.2. Visualization as False Color Images for Human Interpretability and Tumor Classification

The use of a dimensionality reduction technique such as Umap allows us to present hyperspectral data as a false color RGB image. While this inadvertently leads to a loss of information, it enables the visual inspection of the data samples. The focus on the most important features of the spectra further increases the interpretability. In comparison with H&E stained sections a correlation of the learned features with morphological structures could be seen in Figure 2. The three features of the learned dimensionality reduction are shown as a false color image (Figure 2D). The first feature is shown in red and coincides well with the cytoplasm. The green channel shows the second feature and aligns with the background, collagen and the extracellular matrix. The last feature is represented in blue and match with the location of the nuclei.

With this representation resembling natural images as well as medical images from other modalities, we can perform further processing using established ML algorithms.

### 3.3. Deep-Learning-Based Tumor Detection

Using these false color images to fine tune a ResNet image classification network we achieve a sensitivity of 0.40 ± 0.28, specificity of 0.92 ± 0.07, an accuracy of 0.76 ± 0.11 and a F1-Score of 0.73 ± 0.15. The network was already pretrained to recognize a multitude of objects in natural images allowing us to train it to recognize two more classes, tumor and healthy tissue, with few computational resources (Figure 3). One exemplary tissue section with the classification resulting from this approach is shown in Figure 3D. Without the hyperspectral information, using only the RGB images, we achieve a sensitivity of 0.21 ± 0.24, a specificity of 0.94 ± 0.07, an accuracy of 0.76 ± 0.10 and a F1-Score of 0.66 ± 0.14.

By combining the full spectral information with the morphological features of the HSI-data cubes we achieved an average sensitivity of 0.48 ± 0.24, average specificity of 0.89 ± 0.05, an accuracy of 0.76 ± 0.10 and a F1-Score of 0.74 ± 0.14. One exemplary tissue section with the classification results is shown in Figure 3E. Compared to the first ResNet approach where we transformed the HSI data cubes to a more common RGB image format, we observed that the use of the full spectral information increases sensitivity at the expense of specificity. Furthermore, this approach comes at a significantly higher computational cost. All classifications results are summarized in Appendix A.

In Figure 4, we show a detailed zoom on two regions with misclassifications. They were given to a second pathologist for evaluation. According to this reevaluation the false negative classification does not correspond to any histopathological features. The false positive, on the other hand, is in an area with some cell complexes that are ambiguous and cannot be ruled out as tumor tissue purely based on histopathological assessment.

## 4. Discussion

In this study we used HSI for the automated detection of OSCC tumor areas within unstained frozen tissue sections. By showing that we can detect tumor tissue with an accuracy, specificity and sensitivity that is on par with traditional interoperative tumor margin assessment, we introduce hyperspectral imaging as one potential new tool for this task. Both implemented neural networks showed promising results.

Before approaching this task using complex deep learning models we evaluated if the pure spectral information alone would be sufficient to differentiate tumor and non-tumor tissue. Here, we found that a linear support vector machine trained on the individual spectra cannot perform this classification task. While investigating these results we found that the spectral features differ strongly between different measurements due to the manual illumination present in our setup. To overcome this, we used two different deep learning approaches. Deep-learning-based classification algorithms have been shown to be able to handle sample inhomogeneity well as they make use of contextual and morphological features on top of the spectra.

In our first approach we used a combination of two different ML approaches. In the first step we used a ML-based dimensionality reduction technique to turn the 100 channel HSI images into false color images. By employing a ML approach for this step, we kept enough information to then apply a well proven deep-learning-based image classification algorithm and accurately classify tumor and healthy samples. Additionally, the false color images let the human back into the loop by turning high-dimensional HSI images into interpretable false color images that show known morphological features of the samples. This combined approach reaches a sensitivity of 0.40 ± 0.28 and a specificity of 0.92 ± 0.07. We were able to achieve these results with relatively little computational cost since the neural network was pretrained on existing datasets and we only needed to fine-tune it to the HSI classification task. Compared with this, the classification based on the pure RGB images, without any hyperspectral information shows worse classification performance and specifically suffers from a poor sensitivity of just 0.21 ± 0.24. (Specificity: 0.94 ± 0.07, Accuracy: 0.73 ± 0.09, F1-Score: 0.66 ± 0.14).

In our second experiment we skipped the dimensionality reduction and instead trained a neural network specifically design to handle inputs with many channels such as HSI images. While this approach achieved comparable performance with a sensitivity of 0.48 ± 0.24 and an average specificity of 0.89 ± 0.05, this approach has its restrictions in terms of available computational resources and limited sample availability. The network architecture would benefit form more training data and a higher amount of samples.

One interesting result of our study emerged when we decided to have the misclassified samples partially investigated by a second pathologist who had not seen them before. It turned out that while most false negative samples brought no further insight a considerable amount of the false positive classification areas contained ambiguous cell complexes which could not be determined as guaranteed tumor-free cells. This shows the potential for ML to guide the pathologist towards even small areas of interest that would otherwise be given too little attention.

This result combined with the non-destructive, label-free and fast nature of the hyperspectral imaging process shows great potential in future applications. HSI combined with deep learning can serve as a first classification step in interoperative tumor margin assessment and additionally improve the following histopathological evaluation by highlighting potential areas of interest.

Our tile-based approach that divides the tissue section into roughly 100–200 individual regions classifying every single one of them provides sufficient spatial resolution for both applications. This approach could be further improved by training a neural network for segmentation instead of classification. We speculate that accuracy levels comparable to those of an experienced pathologist can be achieved given enough computational resources and an increased number of patients and samples. Given the well-studied adaptability of deep learning methods and the right samples our results should hold true for other tumor entities.

In addition to investigating the presence or absence of tumor, HSI may be useful to detect new molecular signatures. Currently, the detection of molecular alterations at the tumor border are not visible to the pathologists and further tests are needed. The manual analysis is not feasible within the duration of a surgical intervention. Using HSI, the intraoperative molecular analysis of surgical margins may become feasible.

Until now, most deep learning attempts with biological material require further studies and will benefit from more samples because tissue is not a standardizable matter. With more samples the used standard, not domain specific, machine learning approaches could be refined or replaced by more specialized approaches.

Another area for improvement for further research is the image acquisition process as it currently it relies on the combination of two vendor specific programs and manual operation of the hyperspectral camera. This process could be streamlined for further applications.

## 5. Conclusions

In this study, we showed that combining microscopic hyperspectral imaging on unstained fresh-frozen tumor samples with an automated, deep-learning-based tumor classification model should be considered a potential future approach for intraoperative tumor margin assessment. Thanks to the easy-to-use, simple hardware and comparatively fast image acquisition and processing times of around 10 min per slice, hyperspectral imaging could become an additional tool for intraoperative tumor margin detection.

## Figures and Tables

**Figure 1 cancers-15-00213-f001:**
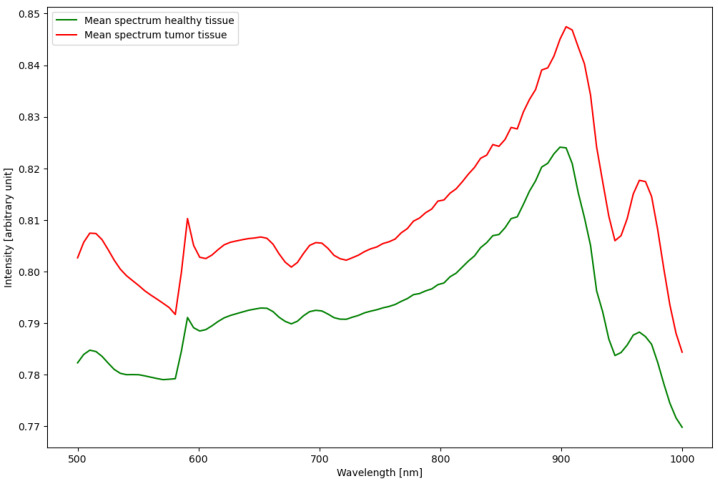
The average spectrum for all pixels labelled either healthy or tumor. A detailed view of each spectrum, including the standard deviation, can be found in the Appendix A.

**Figure 2 cancers-15-00213-f002:**
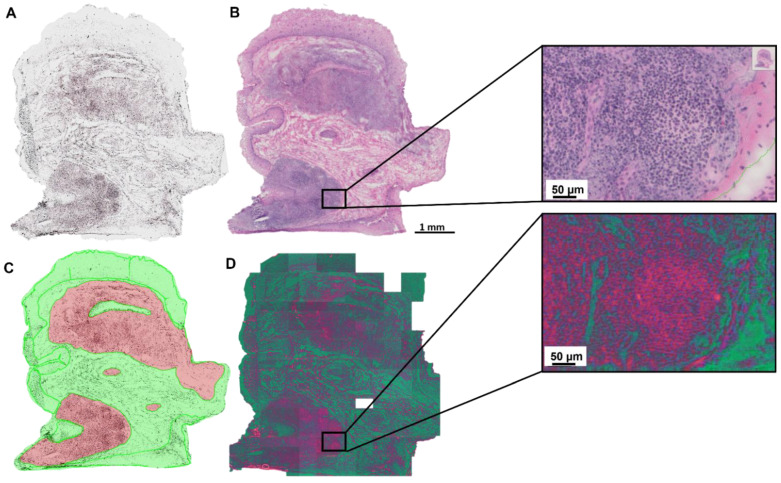
False color image representing the results of the Umap dimensionality reduction. (**A**) unstained tissue section. (**B**) H&E stained tissue section. (**C**) Manual histopathological annotations on H&E-stained tissue section. (**D**) False color image to visualize the results of the dimensionality reduction algorithm. Red feature: cytoplasm, green feature: background, collagen and the extracellular matrix, blue feature: nuclei. The empty white tile in (**D**) was removed from the dataset due to a lack of tissue or poor matching between HSI and microscopic imaging.

**Figure 3 cancers-15-00213-f003:**
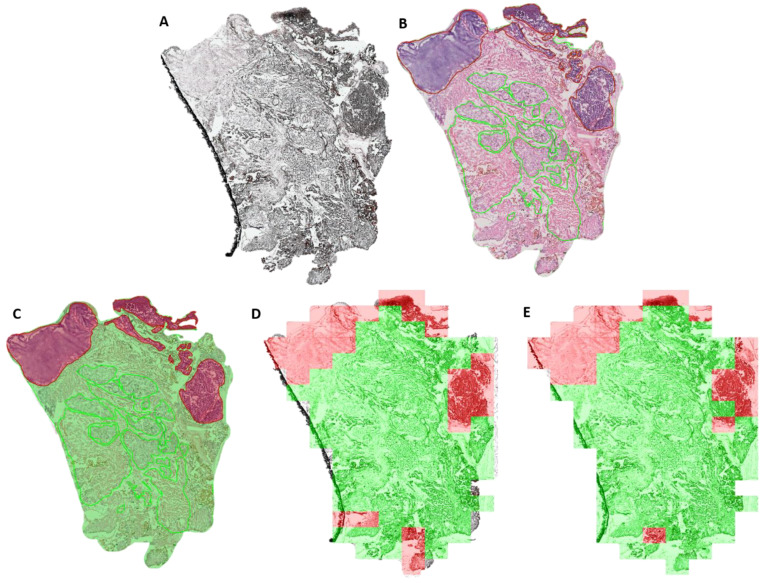
Visualization of the tumor classification results of 3.1.3 and 3.1.4 on an exemplary tissue section. (**A**) unstained tissue section. (**B**) H&E stained tissue section. (**C**) Manual histopathological annotations. (**D**) Tumor classification using a fine-tuned ResNet architecture on false color images. (**E**) Tumor classifications results using a 3D CNN on the full hyperspectral data cubes. Green: healthy tissue. Red: Tumor tissue.

**Figure 4 cancers-15-00213-f004:**
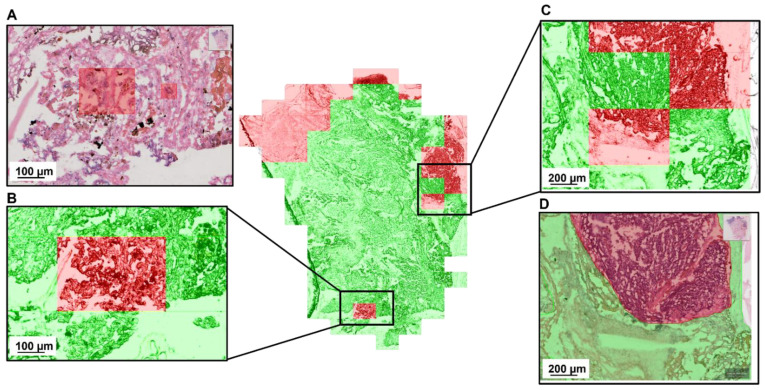
Detailed evaluation of the 3d CNN-based tumor classification with a false positive (**B**) and a false negative (**C**) sample. Both regions were given to a pathologist for reevaluation. The annotation (**D**) was confirmed by the second pathologist. The red regions in (**A**) mark features that the second pathologist labelled as ambiguous.

## Data Availability

The data presented in this study are available on request from the corresponding author. The data are not publicly available due to privacy reasons.

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
