# Peer review of "Intraoperative Assessment of Tumor Margins in Tissue Sections with Hyperspectral Imaging and Machine Learning"

_cancers, 2022, doi:10.3390/cancers15010213_

Round 1

Reviewer 1 Report

The authors are presenting their findings on the classification of unstained OSCC slides based on hyperspectral data from 7 patients and state of the art ML-Methods. Combining HSI and ML for digital pathology is a hot topic and addresses a relevant clinical need. However, the manuscript needs to be improved before publishing. The major concerns are missing standard preprocessing/evaluation for hyperspectral data and comparison with color images from the used RGB-camera.

Title: - The title should be more specific regarding the investigated tissue (intraoperative vs. tissue sections).
Abstract: - The results should be quantified in the abstract and simple summary.
Introduction: - What is the state of the art in OSCC classification with deep learning based on RGB images and the accuracy that was obtained?
General: - The number of patients is very low for the methods 2 and 3.
2.2 HSI Setup: - Please specify the used light source (halogen, LED) for imaging from 500-1000 nm or state the used spectral range.
2.4 Data preprocessing: - Please explain the validation in more detail. How many HSI records were acquired per patient and per slide? Was a balancing of normal and OSCC tiles performed? Does one fold of the validation contains one slide or five (one per patient)? Why did you choose out of five patient and not seven? Leave-one-patient-out should be preferred.
3. Results: - Please provide a figure with the measured mean reflectances and standard deviations for both tissue classes over all patients.
3.1 Line 170-172: - This is a typical issue in HSI and spectral normalization (SNV or Min-Max) is performed to address that. Please state if any spectral preprocessing was performed.
3.2+3.3: Please provide AUC or F1 Score. Results and statistical measures should be provided in additional tables.
3.2: Did the authors perform the fine-tuning and validation of the ResNet with the RGB images from the Axiocam of the same slides too? This would be an interesting comparison and benchmark for the use of HSI in this context.
4. Discussion Line 239: Please elaborate on the "manual illumination" procedure.

Author Response

The authors are presenting their findings on the classification of unstained OSCC slides based on hyperspectral data from 7 patients and state of the art ML-Methods. Combining HSI and ML for digital pathology is a hot topic and addresses a relevant clinical need. However, the manuscript needs to be improved before publishing. The major concerns are missing standard preprocessing/evaluation for hyperspectral data and comparison with color images from the used RGB-camera.

Answer: We gratefully thank the reviewer for the helpful comments. The suggestions were valuable for improving the manuscript’s quality. The feedback helped us to address several key points and strengthen the arguments in our paper especially on the evaluation of the hyperspectral data. In the following we will address the reviewer’s comments in detail.

Title: - The title should be more specific regarding the investigated tissue (intraoperative vs. tissue sections).

Answer: Title was changed to “Intraoperative assessment of tumor margins on tissue sections with Hyperspectral Imaging and Machine Learning”.

Abstract: - The results should be quantified in the abstract and simple summary.

Answer: Quantified results were added to the simple summary and abstract.

Introduction: - What is the state of the art in OSCC classification with deep learning based on RGB images and the accuracy that was obtained?

Answer: We added a paragraph (l. 73 – 77) to sum up the state of the art of deep learning and RGB images.

General: - The number of patients is very low for the methods 2 and 3.

Answer: While it is true that the number of patients is low for a deep learning-based approach, the number of samples is high enough due to our tile-based approach. We added table 1 and 2 in the supplements detailing the number of samples used for each training and discuss the limitations of the comparatively small sample size in chapter 4 (l. 309-316).

2.2 HSI Setup: - Please specify the used light source (halogen, LED) for imaging from 500-1000 nm or state the used spectral range.

Answer: We specified the light source in chapter 2.2 (methods) and implemented a more detailed description (l. 111-112).

2.4 Data preprocessing: - Please explain the validation in more detail. How many HSI records were acquired per patient and per slide? Was a balancing of normal and OSCC tiles performed? Does one fold of the validation contains one slide or five (one per patient)? Why did you choose out of five patient and not seven? Leave-one-patient-out should be preferred.

Answer: We added a more detailed description of the cross validation in chapter 2.4. (l.141-145). Additionally, we added table 1 and table 2 in the supplement stating the exact samples used in each validation step and the number of samples.

Results: - Please provide a figure with the measured mean reflectances and standard deviations for both tissue classes over all patients.

Answer: We added the measured mean intensities for both tissue classes in chapter 3.1 and provide more details in the supplement in Figure 1.

3.1 Line 170-172: - This is a typical issue in HSI and spectral normalization (SNV or Min-Max) is performed to address that. Please state if any spectral preprocessing was performed.

Answer: We did not perform any normalization since we assume that there are potential differences in transmission between different tissue types. To clarify this, we added the relevant information in 2.4 (l. 133-136).

3.2+3.3: Please provide AUC or F1 Score. Results and statistical measures should be provided in additional tables.

Answer: We added the F1 Score and provide a table with all the results in the supplement in Table 3.

3.2: Did the authors perform the fine-tuning and validation of the ResNet with the RGB images from the Axiocam of the same slides too? This would be an interesting comparison and benchmark for the use of HSI in this context.

Answer: We performed the same fine-tuning and validation using ResNet on the RGB images. The results are discussed in 3.3 (l.221-223) and also shown in the supplement in Table 3.

Sensitivity

Specificity

Accuracy

F1-Score

ResNet Finetuning on false color images

0.40 ± 0.28

0.92 ± 0.07

0.76 ± 0.11

0.73 ± 0.15

ResNet Finetuning on RGB images

0.21 ± 0.24

0.94 ± 0.07

0.73 ± 0.09

0.66 ± 0.14

3D CNN

0.48 ± 0.24

0.89 ± 0.05

0.76 ± 0.10

0.74 ± 0.14

Discussion Line 239: Please elaborate on the "manual illumination" procedure.

Answer: We specified the manual illumination procedure in chapter 2.2 (l.111-112) and included a more detailed description.

Reviewer 2 Report

1) Motivation of the propose study can be explained in detail in the introduction. At the end of the introduction, discuss the limitations of the exsisting approaches and how the proposed approach overcomes these limitations

2) Both SVM and CNN have parameters, the optimal performances depends on these parameters. However, the authors didn't discuss anything related to hyperparameter tuning. Without this the experiments are not reproduciable. Authors can read the papers of machine learning and deep learning related in Cancers, mdpi for more details.

3) Authors are suggested to mention the number of data samples used in training, testing, and validation

4) Comaprision of the proposed approach with the exsisting approaches is required.

5) Why SVM is chosen and why not other classifiers

6) Authors are suggested to show the heatmaps or feature visualization using t-SNE

7) Discuss the limitations of the proposed approach and future works

Author Response

Answer: We gratefully thank the reviewer for the helpful comments. The suggestions were valuable for improving the manuscript’s quality. The feedback helped us to address several key points and strengthen the arguments in our paper. In the following we will address the reviewer’s comments in detail.

  • Motivation of the propose study can be explained in detail in the introduction. At the end of the introduction, discuss the limitations of the exsisting approaches and how the proposed approach overcomes these limitations.

Answer: We discuss the limitations of the existing approaches at the beginning of the introduction and have now added further details to this discussion (l. 58-62; l. 73-77).

  • Both SVM and CNN have parameters, the optimal performances depends on these parameters. However, the authors didn't discuss anything related to hyperparameter tuning. Without this the experiments are not reproduciable. Authors can read the papers of machine learning and deep learning related in Cancers, mdpi for more details.

Answer: We did not perform hyperparameter tuning in this study. Since we mainly focused on showing the feasibility of our approach, we decided to use common/standard hyperparameters for the used models. The applied hyperparameters can be found in the materials and methods chapter (l. 159-161; 167-168; 175-176). Nevertheless, in further studies including more patients and samples a hyperparameter optimization would be beneficial.

  • Authors are suggested to mention the number of data samples used in training, testing, and validation

Answer: We added a table in the supplements (Table 2) showing the exact number of samples in each fold of the 6-fold-crossvalidation.

  • Comaprision of the proposed approach with the exsisting approaches is required.

Answer: We added a more detailed description of the existing approaches in the introduction (l. 58-62; l.73-77).

  • Why SVM is chosen and why not other classifiers

Answer: We chose SVM, since it is one of the most used classical machine learning classifier. Additionally, it is a classifier that has seen regular use in hyperspectral image classification. We added additional details and references in chapter 2.6.1 (l.159-161).

  • Authors are suggested to show the heatmaps or feature visualization using t-SNE

Answer: We performed feature visualization using UMAP which generally produces results comparable to t-SNE. The resulting feature visualization was, in our opinion, not helpful and is therefore not part of the final manuscript.

  • Discuss the limitations of the proposed approach and future works

Answer: We added a more detailed discussion of the limitations (l.309-312) and speculate that in this setting the use of a specialized network architecture could further increase the classification performance.